# Oncotherapeutic Strategies in Early Onset Colorectal Cancer

**DOI:** 10.3390/cancers15020552

**Published:** 2023-01-16

**Authors:** Mary O’Reilly, Anna Linehan, Aleksandar Krstic, Walter Kolch, Kieran Sheahan, Des C. Winter, Ray Mc Dermott

**Affiliations:** 1Centre for Colorectal Disease, St. Vincent’s University Hospital, Elm Park, D04 YN26 Dublin, Ireland; 2Systems Biology Ireland, School of Medicine, University College Dublin, Belfield, D04 V1W8 Dublin, Ireland; 3Conway Institute of Biomolecular & Biomedical Research, University College Dublin, Belfield, D04 V1W8 Dublin, Ireland

**Keywords:** colorectal cancer, early onset, therapeutics, treatment, onco-therapeutics

## Abstract

**Simple Summary:**

Early onset colorectal cancer (defined as colorectal cancer in those less than 50 years of age) is increasing in incidence worldwide. There are many risk factors for the development of early-onset colorectal cancer, including genetic predisposition syndromes, the disruption of gut microbiota, obesity, and inflammatory bowel disease. Due to a lack of awareness of this condition in both the medical and general community, there is often a delay in reaching this diagnosis; this leads to worse outcomes than are present among their older counterparts. This is a cohort with specific physical, emotional, social, and financial needs. The aim of this review is to discuss the etiology and treatment strategies for this patient group.

**Abstract:**

Early onset colorectal cancer (EOCRC), defined as colorectal cancers in patients aged less than 50 years, is becoming an increasingly common issue, globally. Since 1994, the incidence of this condition has been rising by 2% annually. Approximately one in five patients under 50 years of age diagnosed with colorectal cancer have an underlying genetic predisposition syndrome. The detection of cancer among the other 80% of patients poses a considerable task, as there is no family history to advocate for commencing early screening in this group. Patients with EOCRC have distinct social, spiritual, fertility, and financial needs from their older counterparts that need to be addressed. This review discusses the risk factors associated with the development of EOCRC and current best practice for the management of this disease.

## 1. Introduction

Colorectal cancer is the third most common cancer among either sex in the United States and also the third most common cause of cancer-related mortality [1]. Early onset colorectal cancer (EOCRC) now accounts for 1 in 10 diagnoses of this cancer subtype [2]. The distinction between EOCRC and late-onset colorectal cancer (LOCRC) is not clear. However, it is largely agreed that age 50 is an acceptable arbitrary cut-off value, as this is the age that many countries begin to screen for colorectal cancer (CRC). The NCRI database also shows that the incidence of CRC among those under 50 is rising at an alarming rate [3]. This trend of increased incidence of EOCRC is seen across the globe, while the incidence of LOCRC is steadily declining. A recent publication analyzing the worldwide trends of EOCRC showed that the incidence of CRC increased by more than 100% from 95,737 in 1990 to 226,782 in 2019 [4].

The issue of the increasing incidence of EOCRC brings many challenges and clinical dilemmas. The first issue is trying to understand why the incidence of the disease is increasing at such a high rate in our youth. Since 1994, CRC incidence in individuals younger than 50 years has been increasing by 2% per year, whilst stability, and even a decline, in the later onset group has been seen [5]. The decline in incidence seen in the later onset cohort is thought to be due to an increase in screening for colorectal cancer. Screening in the later onset cohort acts to identify precancerous polyps or early-stage adenomas. The purpose of this review is to summarize current guidance with regard to the treatment strategies of EOCRC.

## 2. Risk Factors for the Development of Colorectal Cancer

CRC is a heterogeneous disease, with many risk factors increasing the likelihood of developing the disease. A history of inflammatory bowel disease increases the risk of the development of CRC [6]. Other risk factors for the development of CRC include smoking, alcohol consumption, a diet high in red and processed meats, diabetes mellitus, physical inactivity, raised BMI (body mass index), and early exposure to antibiotics [7]. Risk factors for development of CRC are summarized in Figure 1. In the EPIC cohort of almost 350,000 individuals, those who adhered to five healthy lifestyle factors (normal BMI, recommended amount of physical activity, non-smoking, mild alcohol consumption, and healthy diet) had a hazard ratio for the development of CRC of 0.63 (95% CI, 0.54–0.74) compared with those who adhered to 1 or none of the factors [8]. The 2010 global burden of disease study identified a diet lacking in milk and a diet lacking in whole grains to be the main risk factors for the development of EOCRC worldwide [4]. All of the risk factors described share a common endpoint of the altered constitution of the gut microbiota. Intestinal dysbiosis is understood to play a pivotal role in carcinogenesis by disrupting the anti-inflammatory effects of the gut microbiome [9]. Many dietary compounds, such as heterocyclic amines and polycyclic aromatic hydrocarbons, directly exert genotoxic effects on the gut as the barrier function of the microbiome is disrupted. An interesting study demonstrated, over generations of mice, that a diet of high fat and simple carbohydrates resulted in an altered microbiome with reduced diversity [10]. These changes cause an increased incidence of inflammatory and malignant intestinal disease in murine models [11].

## 3. Cancer Predisposition Syndromes

Hereditary cancer syndromes have an incidence rate of less than 5% in the general population [12] and are an important cause of CRC. The incidence is higher in younger patients, with approximately 20% of those under 50 who are diagnosed with colorectal cancer being affected by hereditary cancer syndromes [5]. The most common hereditary cancer syndrome is Lynch syndrome; it accounts for 2% to 4% of all CRC cases [13] and approximately one-third of the CRC cases in patients younger than 35 years [14]. It is inherited in an autosomal dominant manner. This hereditary syndrome results from germline mutations in DNA mismatch repair (MMR) genes (MLH1, MSH2, MSH6, and PMS2) [15], as seen in Figure 2. Its presence is usually detected by analyzing for microsatellite instability (MSI), which results from MMR deficiency and is detected as alterations in the length of repetitive DNA sequences in the tumor tissue, caused by the insertion or deletion of repeated units [16]. If immunohistochemistry shows that MLH1 expression is absent in the tumor, then it is necessary to test for mutations in the BRAF gene. If a BRAF mutation is present, it is not a case of Lynch syndrome [17]. A mutation of BRAF indicates that MLH1 expression is down-regulated through the somatic methylation of the promoter region of the gene [18]. Another way to determine this is by testing for MLH1 promoter methylation. A large, retrospective, international, multicenter, observational study found 25% of EOCRCs to demonstrate MSI [19].

Other hereditary cancer syndromes, such as familial adenomatous polyposis (FAP), also increase the risk of developing colorectal cancer. FAP is inherited in an autosomal dominant manner. Overall, 80% of patients with FAP have a germline mutation in the adenomatous polyposis coli (APC) gene [20]. A subset of people with FAP has a biallelic mutation of the MYH gene [21]. Patients with FAP tend to develop more than 100 colorectal adenomas. If left without treatment, the majority will develop colorectal adenocarcinoma by 40 years of age. A variant is attenuated FAP, where patients develop 10 to 100 colorectal adenomas. A clear genotype–phenotype relationship exists: APC mutations between codons 1445 and 1578 are associated with an increased risk of desmoid tumors—patients with attenuated FAP usually have mutations at the 5′ (proximal to codon 1517) or the 3′ end (distal to codon 1900) of the APC gene. Patients who have FAP where an identifiable mutation cannot be found are at a higher risk for more aggressive disease [22].

## 4. Diagnostic Delays

The initial investigation for patients undergoing screening or experiencing symptoms suggestive of colorectal cancer is a colonoscopy; this will confirm the diagnosis pathologically.

Once the diagnosis is confirmed, baseline computed tomography (CT) of the chest, abdomen, and pelvis and a carcinoembryonic antigen (CEA) test are the standard studies for the investigation of colorectal cancer. Magnetic resonance imaging (MRI) of the pelvis may be requested by surgeons to decipher whether a patient with rectal cancer is a surgical candidate and to assist with planning the surgery. Occasionally, positron emission tomography (PET) scanning may be useful if metastatic disease cannot be ruled out by CT. The most important prognostic colorectal cancer indicator is the pathological stage at presentation. The tumor stage is calculated using TNM staging.

The fact that these patients typically seek medical attention much later than their older counterparts is a significant challenge. Studies have shown that painless bleeding, which is often dismissed as hemorrhoids, can occur up to 3 years before other CRC symptoms [23]. A sense of invincibility in young adults, combined with a lack of medical insurance in the United States, contributes to delayed presentation [24]. CRC diagnosis in younger patients is usually determined, on average, 6 months after onset of symptoms [25]. Society still views this disease as a condition of older age. Therefore, hesitancy and misdiagnosis on the part of both the patient and physician, who expect the symptoms to be related to a benign disease process, are common. Due to the delay in diagnosis, there is a higher proportion of patients presenting with metastatic disease compared to their older counterparts. Studies have shown that up to 76% of patients younger than 30 years of age present with stage III or IV disease [26], compared with less than 50% of patients over 50 years of age [27]. Increased awareness of the incidence of colorectal disease among the young is needed in both the medical and wider community.

## 5. Treatment Strategies

Another challenge in the treatment of colorectal cancer is the nonuniform response to anti-cancer drugs and resistance to drugs to which the cancer was once susceptible. This review will discuss treatment strategies for patients with EOCRC. While many of the principles for treating cancer of the colon and the rectum are similar in the EOCRC cohort compared with the LOCRC cohort, there are some differences. The EOCRC cohort generally has fewer comorbidities and higher performance status, meaning that they tend to be treated more aggressively with chemotherapy and radiotherapy by oncologists [28]. The need to discuss fertility risks and counseling regarding family planning presents itself more frequently in the EOCRC cohort. There are inconsistencies in the literature as to whether this aggressive approach translates to an improved prognosis, with some data reporting a similar or improved survival [29,30] and others reporting a worse prognosis in EOCRC patients [31]. A multimodal, patient-tailored approach—combining pharmaceutical agents, such as chemotherapy and immunotherapy, radiotherapy, and surgery—may be essential to improve morbidity and mortality from this disease. Much work is underway to develop drugs to target the driver mutations of EOCRC. This personalized medicinal approach will reduce the number of long-term side effects of chemotherapy that this group is exposed to.

In order to divide colorectal cancer into subtypes to predict responsiveness to antineoplastic drugs, many attempts to identify genetic targets have been made [32]. Unfortunately, the identification of molecular markers to guide therapy selection in colorectal cancer has not been plentiful. The targets currently being explored include KRAS mutations. A recent phase I/II study showed the benefit of using targeted therapy in patients with GI cancers harboring KRAS G12C mutations [33]. HER-2, NTRK, and RET driver mutations can be targeted with drugs, but these are rarely seen in colorectal populations. Those with microsatellite instability should be treated with immunotherapy [34]. Genes that lead to the development of colorectal cancer are either oncogenes that have developed enhanced functioning or tumor suppressor genes that have lost their function. The Cancer Genome Atlas has identified 32 frequently mutated somatic genes in colorectal cancer, of which, APC, TP53, KRAS, NRAS, PIK3CA, FBXW7, SMAD4, TCF7L2 CTNNB1, SMAD2, FAM123B (also known as WTX), and SOX9 were the most commonly mutated [35]. Figure 3 shows common genetic mutations seen in EOCRC. PIK3CA is the only mutated gene in this list that can be targeted with an FDA-approved drug. A study by Yuan et al. used multiomics to identify genetic susceptibility genes for colorectal cancer, and they were successful in showing 66 susceptibility genes and their loci [36]. As part of their study, they combined the regression results of the expression of quantitative trait loci analyses from three large studies, the Genotype-Tissue Expression, Cancer Genome Atlas, and Colonomics projects, with their meta-analysis. This enabled them to identify 18 additional susceptibility genes in CRC which were not previously described. Xu et al. [37] sought to identify new genetic drivers associated with CRC. They identified two new genes associated with CRC, LRRC26 and REP15, which also acted as prognostic biomarkers. Studies have shown that somatic copy number alteration (SCNA) is one of the most common structural mutations in colorectal cancer [38]. SCNA genes are regarded as driver genes for cancer development [39]. A number of SCNA genes are showing potential for use as prognostic markers for patients with colorectal cancer. Studies have shown that a high copy number of mitochondrial DNA can indicate a poor prognosis in advanced colorectal cancer, suggesting that metabolic changes are associated with more aggressive forms of the disease [40]. Further studies to ensure consistency and reproducibility are needed.

Due to the difficulty of developing a classification relying on genomic alterations, an expression-based classification was proposed. In 2015, a new stratification of molecular subtypes was developed by an international consortium that integrated data from six different transcriptome-based classifiers [41]. The four identified consensus molecular subtypes (CMS) show the individual activation levels of cancer pathways and distinct genomic alterations: CMS1 (MSI immune), microsatellite unstable, and hypermutated; CMS2 (canonical), epithelial, chromosomally unstable, marked MYC- and WNT-signaling activation; CMS3 (metabolic), epithelial, with evident metabolic dysregulation; and CMS4 (mesenchymal). It has been shown that the CMS subtypes have predictive and prognostic abilities—an example of this was improvements in overall survival that were associated with the use of Bevacizumab compared with Cetuximab in CMS1 [42].

## 6. Chemotherapy Regimens Used in EOCRC

5-Fluorouracil is the first-line chemotherapeutical agent used for CRC. It is usually used in combination with other chemotherapeutical agents. It is administered in two ways: either by continuous infusion via access to central venous circulation or orally in the form of Capecitabine. It works as an antimetabolite to prevent cell proliferation. It primarily inhibits the enzyme thymidylate synthase, thereby blocking the thymidine formation required for DNA synthesis [43]. Other drugs commonly used in CRC include Oxaliplatin, a platinum drug that inhibits DNA synthesis [44], and Irinotecan, which works by inhibiting topoisomerase I [45].

## 7. Treatment of Metastatic EOCRC

Metastatic colorectal cancer with no potential for surgical resection cannot be cured. The necessary considerations when making treatment decisions will include the tumor characteristics, the patient’s performance status and comorbidities, the treatment schedule, expected toxicities, and the mutational profile of the tumor. The aim of treatment in this setting is to improve quality of life and prolong life expectancy. The current treatment options include various drugs, which can be used in combination or as single agents. First-line systemic treatment options for those who are microsatellite stable (MSS) and do not have a druggable driver mutation include FOLFOX (continuous infusion of 5-FU and Oxaliplatin), CAPEOX (Capecitabine and Oxaliplatin), FOLFIRI (continuous infusion 5-FU and Irinotecan), and FOLFOXIRI (continuous infusion of 5-FU, Oxaliplatin, and Irinotecan) [46]. Large, randomized control trials and meta-analyses have not selected one of the above chemotherapy combinations to be superior to the others [47,48]. Oxaliplatin can cause peripheral neuropathy and thrombocytopenia. Irinotecan can cause diarrhea and lymphocytopenia.

## 8. Additional Targeted Agents in Metastatic EOCRC

All patients with metastatic disease should have their tumors assessed for mutations in KRAS, NRAS, and BRAF. This testing can be conducted on the primary colorectal tumor or the metastasis, as the results are similar in both specimen types [49]. Bevacizumab is a humanized monoclonal antibody that counteracts the effects of vascular endothelial growth factor (VEGF), which is important in tumor angiogenesis. A number of metanalyses have shown that the use of Bevacizumab in combination with chemotherapy in a first-line setting confers a clinical benefit [50,51]. Cetuximab and panitumumab are monoclonal antibodies directed against the epidermal growth factor receptor (EGFR). Cetuximab is a chimeric monoclonal antibody, whereas Panitumumab is a completely human monoclonal antibody. Patients with a known KRAS mutation (exon 2, 3, 4) or NRAS mutation (exon 2, 3, 4) should not be treated with either anti-EGFR therapy cetuximab or panitumumab [52]. A BRAF V600E mutation makes a response to panitumumab or cetuximab highly unlikely unless given together with a BRAF inhibitor [53]. There is evidence for use of EGFR inhibitors in RAS wild-type, left-sided primary tumors (splenic flexure to rectum). The phase III CALGB/SWOG 80,405 trial [54] assessed the use of EGFR inhibitors vs. VEGF in combination with chemotherapy in a first-line setting. They found that RAS wild-type, right-sided tumors (caecum to hepatic flexure) had longer overall survival if treated with bevacizumab than if treated with cetuximab in a first-line setting. Patients with RAS wild-type, left-sided primary tumors (splenic flexure to rectum) had a longer overall survival if they received treatment with cetuximab rather than bevacizumab.

## 9. Manipulation of The Immune System to Treat EOCRC

The recent advent of immunotherapy commenced a renaissance of oncotherapeutics in EOCRC. In CRC, the immune microenvironment is crucial for disease development, response to treatment, and prognosis [55]. Those CRCs that demonstrate microsatellite instability (MSI) have a higher tumor mutational burden [19]. Approximately 5% of all metastatic colorectal cancer will be MSI high [56]. MSI high status is a biomarker that indicates the response to immunotherapy. These patients should receive a PD-1 inhibitor alone or in combination with a CTLA-4 inhibitor [57]. Much work has been done to develop an ‘immunoscore’, which is a classification system to predict responses to immunotherapy based on CD3 and CD8 lymphocyte populations in the tumor core and invasive margin. It is not currently used in clinical practice [58]. The tumor microenvironment in MSS disease is one which comprises cancer-associated fibroblasts and pro-tumorigenic macrophages, which cause the suppression of T cells and promote tumor development [59]. Immunogenic cell death can be triggered by therapies, such as radiation or chemotherapy [60], and it is hypothesized that this may render the tumor more susceptible to immunotherapy.

POLE and POLD1 are genes that encode DNA polymerases with proofreading activity [61]. Mutations in these genes result in large numbers of single nucleotide variants. Colorectal cancers driven by somatic POLE mutations occur in 1–2% of colorectal cancers; POLD1 mutations are very uncommon [62]. POLE mutant colorectal cancers carry a favorable prognosis [63]. They feature excellent responses to immune checkpoint blockades [64] due to the high mutational burden, which triggers a strong neoantigen response [65]. Temko et al. [66] analyzed endometrial and colorectal carcinomas and colorectal adenomas to assess how early in tumorigenesis the POLE mutation occurs and the immune response associated with the mutation. They found that the T cell infiltrate in POLE mutant cancers was the same as the infiltrate in the precursor lesions, and the absolute number of neoantigens was much higher than in other cancers. From this data, it could be inferred that somatic POLE mutations are early events in the cancers in which they occur. Given that these cancers are relatively common and appear to respond to checkpoint inhibition, consideration should be given to checking MSS EOCRC for somatic POLE mutations.

## 10. Treatment of Early-Stage EOCRC

Surgical resection is the main treatment modality for localized early-stage colon cancer. Patients with a bulky nodal disease on imaging or clinical T4 tumors should be considered for neoadjuvant chemotherapy with fluoropyrimidine-based chemotherapy to decrease the size of the tumor prior to resection, based on evidence from the FOxTROT study [67]. This study showed that neoadjuvant chemotherapy was associated with a decrease in incomplete resections (5% vs. 10%; *p* = 0.001). The status of the resection margins as well as the number of removed nodes are very important determinants of prognosis.

Adjuvant therapy can augment the chance of a cure in patients with high-risk colon cancer. All patients with a node-positive disease should receive chemotherapy with a 5-FU element (i.e., FOLFOX or CAPEOX) for 3–6 months or 6 months of capecitabine. There is no consensus over which regimen is superior [68]. Chemotherapy should be considered in patients who have a node-negative disease with a T4 tumor or a T3 tumor with disease features that indicate there is a high risk of recurrence (i.e., poorly differentiated/undifferentiated histology; lymphatic/vascular invasion; bowel obstruction; <12 lymph nodes examined; perineural invasion; localized perforation; close, indeterminate, positive margins; or tumor budding) [69].

Universal testing for MMR or MSI is recommended for all EOCRC patients. The literature suggests that a deficiency in MMR protein expression or MSI-high (MSI-H) tumor status may be a predictive marker of decreased benefit, and possibly even a detrimental impact, from adjuvant therapy with fluoropyrimidine in patients with stage II disease. Thus, in those patients, adjuvant chemotherapy should be avoided [70].

For low-risk rectal cancers (T1–T2 with no positive nodes and no high-risk features), the recommended practice is transabdominal resection with total mesorectal excision. Neoadjuvant or adjuvant therapy is not recommended in this setting [71]. If the tumor is upstaged pathologically, then the patient should undergo adjuvant chemotherapy. For any tumors that are T3 or T1–2 tumors with positive nodes, the literature shows that there are benefits to treating the tumor preoperatively with chemoradiotherapy and chemotherapy upfront [71]. This is called total neoadjuvant therapy. A number of randomized trials have established the benefit of adding chemotherapy (most often 5-FU/leucovorin or capecitabine) to radiotherapy for the treatment of localized rectal cancer. The use of concurrent chemoradiation is thought to increase local radiotherapy sensitization and the systemic control of the disease (i.e., increased pathological responses and the eradication of micrometastases) [72]. Preoperative chemoradiotherapy can increase the rates of pathologically complete responses and sphincter preservation [73].

## 11. Oligometastatic Disease

Approximately 50% of patients diagnosed with CRC develop colorectal metastases [74,75], with more than 80% of these patients developing unresectable metastatic liver disease [76]. A subset of these patients will be candidates for surgical resection with curative intent. Meta-analyses report a 5-year overall survival rate as high as 71% following the resection of a solitary liver lesion [77]. Determining which patients are surgical candidates is essential; all diseased tissue should be removed with clear surgical margins, and a sufficient quantity of liver tissue should remain to allow for adequate liver function. Some data have shown that there is a higher rate of hepatic resection among younger patients [78]. However, a review carried out in the English NHS comparing the rates of hepatic resection in EOCRC to LOCRC for all cancers diagnosed between 2014 and 2018 did not show a significant difference between the two groups [79]. Another study showed that older populations can also have favorable outcomes, and that age alone should not be used as the deciding factor for hepatic resection [80]. An interesting study by Parisi et al. assessed overall survival after liver resection for colorectal metastasis versus non-colorectal metastasis and found the colorectal group had an overall survival time of 54 months vs. 32 months in the non-colorectal metastasis group [81]. Those who are not surgically resectable at the outset can be treated neoadjuvantly in an attempt to shrink the tumor and render the patient resectable. There are a number of 5-FU-based regimens, such as FOLFIRI, FOLFOX, or FOLFOXIRI, in combination with anti-epidermal growth factor receptor (EGFR) inhibitors, that have been successful in converting unresectable disease to resectable disease [82,83]. It is common practice that even patients who are resectable candidates from the outset should complete peri-operative chemotherapy. There is a debate as to which regimen should be followed and whether neoadjuvant or adjuvant therapy is superior. It is essential that patients with resectable disease do not miss the opportunity to proceed to surgery if their disease progresses whilst on neoadjuvant chemotherapy. In total, patients should not have more than 6 months of perioperative chemotherapy. The phase III Tribe study assessed the efficacy of upfront treatment with Bevacizumab combined with FOLFOX, FOLFIRI, and FOLFOXIRI [84]. The conclusion was that the three regimens are equivalent and can be used in the neoadjuvant setting prior to surgery. In patients with dMMR, immunotherapy can be considered. In this instance, there is not yet any clinical trial data, but there are a number of case studies that have reported benefits from immunotherapy before hepatic resection [85].

In the same way that hepatic resection can be used with curative intent, highly selected patients with isolated pulmonary metastasis can undergo resection in a similar manner. One analysis showed a 3-year overall survival rate of 78% [86]. There is no clear consensus on the use of local therapies for non-resectable metastatic disease. Transcatheter arterial chemoembolization, Yttrium-90 microsphere radioembolization, and hepatic arterial infusion chemotherapy have similar levels of efficacy in patients with unresectable colorectal hepatic metastases [87]. There is debate about the timing of the use of the mentioned modalities as well as the use of these modalities.

## 12. Future Perspectives for the Treatment of Early Onset Colorectal Cancer

There are a number of emerging strategies that have the potential to revolutionize the treatment of EOCRC. Liquid biopsies hold the promise to become a gold standard method of diagnosing and monitoring those with colorectal cancer in the near future. The wealth of information that liquid biopsies can provide has numerous applications, including diagnosis, prognosis, therapeutic response prediction, and the detection of relapse in those who were treated with curative intent. The liquid biopsy takes the form of a small-volume (approx. 10 mL) venous blood sample, which can be collected in a low-cost, minimally invasive manner [88]. After diagnosis and treatment commencement, it can be used in conjunction with tissue biopsy to assess responses to treatment and disease relapse. ctDNA is a tumor-specific free DNA fragment released during apoptosis from the primary tumor or associated metastatic deposits, thus allowing it to serve as a biomarker for diagnosis, prognosis, and prediction [89]. It is thought that ctDNA is released by all tumor sites, including metastasis, and therefore, may provide invaluable information with regard to tumor heterogeneity. Tumor-specific mutations allow the tumor DNA to be differentiated from other non-specific free DNA that originates from normal cells. There are two ways in which ctDNA is currently used for therapeutics [90]. The first method is using a targeted approach to detect specific mutations in genes that are actionable, such as NTRK and ERBB2. The other method is an untargeted approach, where whole genome sequencing is performed on the DNA.

As discussed earlier in this review, targeted therapies for the treatment of EOCRC have the potential to effectively treat the condition whilst decreasing therapy-associated toxicities. A promising therapeutic target in colorectal cancer is the enzyme nicotinamide N-methyltransferase (NNMT), which is upregulated in this malignancy and contributes to its aggressiveness [91]. A number of NNMT inhibitors are already available and could be tested for colorectal cancer management and also for enhancing chemotherapeutic efficacy [92,93,94].

Finally, changes to the gut microbiome may be a cause or indicate the development of CRC. Much work is ongoing to identify techniques to prevent and treat CRC. One area that is under scrutiny is the identification of long non-coding RNAs (lncRNAs), which are RNA segments longer than 200 nucleotides that do not have the ability to code for proteins. LncRNAs can be used to diagnose and recognize the progression, recurrence, and chemoresistance of CRC. LncRNAs have the potential for exploitation to develop new strategies that can assist in diagnosis, prognostication, and drug development for colorectal cancer [95,96].

## 13. Conclusions

EOCRC has become a terrifyingly prevalent medical illness in the last two decades. This review has discussed the incidence, etiology, and treatment strategies for EOCRC. Although there are numerous hypotheses attempting to explain the underlying causes of this cohort effect, we do not completely understand the recent dramatic increase in incidence. It is likely that genetics, lifestyle factors, and the microbiome all influence this condition. Large-scale clinical trials have improved the prognosis of patients at the population level by devising protocols based on data collected for the cohort as a whole. However, we know that significant intra-patient and intra-tumor heterogenicity exists. This is especially true for EOCRC, as we do not fully understand the pathogenesis of the disease. It is clear that standard, first-line regimens should not serve in a ‘one size fits all’ manner. The journey toward precision oncology is a critical step in improving outcomes for individual patients. However, developing tailor-made treatment plans for patients based on the tumor’s perceived vulnerabilities to specific agents remains challenging. The diversity across human cancers and limited knowledge of drug–tumor interactions remain as barriers. Further multiomic studies of colorectal cancer hold the key to unlimited access to precision oncology. As previously suggested, it appears that EOCRC behaves as a distinct clinical and molecular entity to LOCRC. Additionally, these patients have complex physical, spiritual, sexual, and psychosocial needs, which may require a multidisciplinary approach involving the general practitioner, surgeon, radiation oncologist, medical oncologist, psychologist, and fertility specialist. Continued research to obtain a better understanding of the reasons why these patients are developing the disease will enable us to devote more resources to the prevention and cure of the disease.

## Figures and Tables

**Figure 1 cancers-15-00552-f001:**
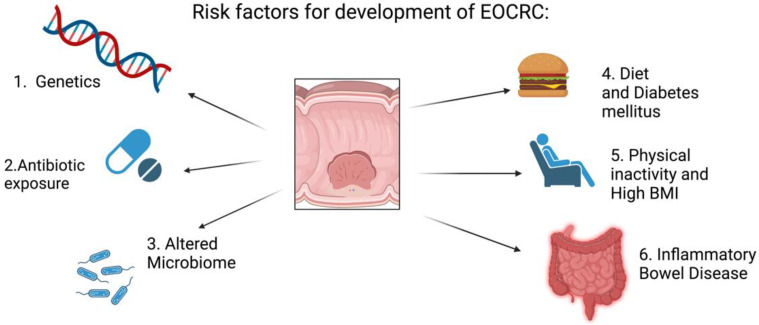
Risk factors for the development of EOCRC. Figure developed using Biorender.com.

**Figure 2 cancers-15-00552-f002:**
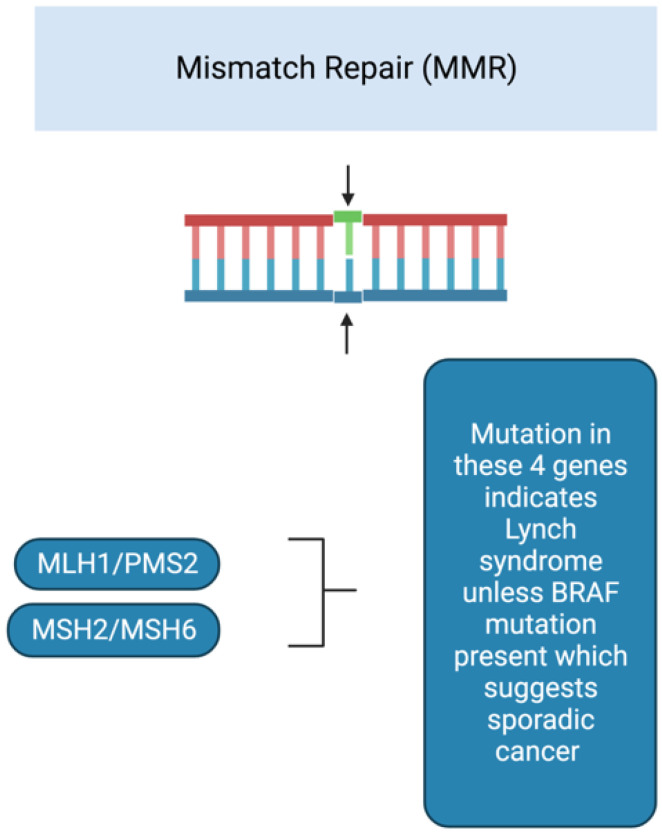
Genes involved in the development of Lynch syndrome. Figure developed using Biorender.com.

**Figure 3 cancers-15-00552-f003:**
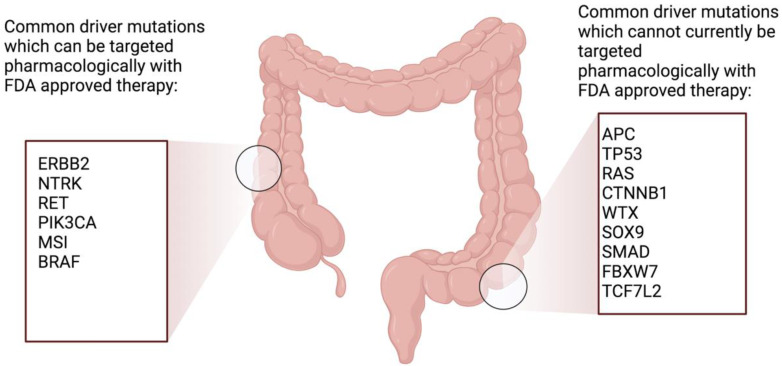
Common genetic mutations seen in EOCRC. Figure developed using Biorender.com.

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
