# Peer review of "Oncotherapeutic Strategies in Early Onset Colorectal Cancer"

_cancers, 2023, doi:10.3390/cancers15020552_

Round 1

Reviewer 1 Report

The manuscript “Oncotherapeutic Strategies in Early Onset Colorectal Cancer” is a concise review article regarding the epidemiology, risk factors, prevention, diagnosis, and treatment of colorectal cancer. The manuscript is well written and it can be of interest for the readers. However, there are some important concerns that must be addressed in order to consider the manuscript suitable for publication:

1.       The figure 1 should be improved. It summarizes only few of the risk factors described in the paragraph. It should include diabetes mellitus, physical inactivity and high BMI.

2.       The paragraph Diagnostic delays should be improved providing information about the current diagnostic staging and prognostic factors.

3.       The manuscript lacks of a paragraph regarding the future perspectives for treatment of Early Onset Colorectal Cancer, summarizing the most recent research directions. For instance, a promising therapeutic target in Colorectal Cancer is the enzyme nicotinamide N-methyltransferase (NNMT) which is upregulated in this malignancy and contributes to its aggressiveness (PMID: 36139012). A number of NNMT inhibitors are already available and could be tested for Colorectal Cancer management, also for enhancing the chemotherapeutic efficacy (PMID: 34572571; PMID: 34704059; PMID: 34424711). Other promising perspectives are associated to the regulation of long non-coding RNAs (e.g. PMID: 36497293; PMID: 36482019; PMID: 36465270).

Minor

Line 132 and 225: the dot is placed before the reference.

Author Response

Thank you very much for your thorough review. 

I have amended the manuscript to include your suggestions:

  1. Figure 1 has been amended to include more risk factors.
  2. Diagnostic delays has been improved, and I have provided information about the current diagnostic staging and prognostic factors.
  3. I have added a new section on the future perspectives for treatment of Early Onset Colorectal Cancer and included the interesting references you suggested. 

Many thanks for your time and expertise. 

Reviewer 2 Report

The manuscript is interesting. the authors present a review that seeks to highlight the best strategies and knowledge in terms of etiology, diagnosis and treatment of early onset colorectal cancer. The manuscript has some biases that should be corrected. It is necessary to improve the English language and to improve the structure of the manuscript it would be advisable to insert tables to schematize the data and also to elaborate a statistical analysis. Furthermore, the references need to be improved (for example, it could be useful to include in chapter 4 - Diagnostic Delays the citation of the manuscript: Analysis of long-term results after liver surgery for metastases from colorectal and non-colorectal tumors: A retrospective cohort study. International Journal of Surgery, 2016, 30, pp. 25–30; and in chapter 7- treatment of metastatic EOCRC the citation relating to the manuscript: Surgery for colorectal cancer in the elderly: a comparative analysis of risk factor in elective and urgency surgery. Aging Clinical and Experimental Research, 2017, 29(1), pp. 65–71

Author Response

Thank you very much for your time and expertise in reading this manuscript. I have edited the English language to make it more comprehendible.

I have updated the references and included the excellent studies you suggested as follows:

Analysis of long-term results after liver surgery for metastases from colorectal and non-colorectal tumors: A retrospective cohort study. International Journal of Surgery, 2016, 30, pp. 25–30

  • I added this very interesting reference to the section on treatment of oligo metastatic disease.

Surgery for colorectal cancer in the elderly: a comparative analysis of risk factor in elective and urgency surgery. Aging Clinical and Experimental Research, 2017, 29(1), pp. 65–71

  • I also added this very informative paper to the section in the paper regarding oligometastatic disease and heapatic resection.

Round 2

Reviewer 1 Report

The manuscript has been improved and can be published.